# 3D-Printed Antibacterial Scaffolds for the Regeneration of Alveolar Bone in Severe Periodontitis

**DOI:** 10.3390/ijms242316754

**Published:** 2023-11-25

**Authors:** Konstantinos Theodoridis, Athanasios S. Arampatzis, Georgia Liasi, Lazaros Tsalikis, Panagiotis Barmpalexis, Dimitrios Christofilos, Andreana N. Assimopoulou

**Affiliations:** 1Laboratory of Organic Chemistry, School of Chemical Engineering, Aristotle University of Thessaloniki, 54124 Thessaloniki, Greece; kostastheod82@yahoo.gr (K.T.); arampatzisa@cheng.auth.gr (A.S.A.); georgiasl@cheng.auth.gr (G.L.); 2Natural Products Research Center of Excellence (NatPro-AUTH), Center for Interdisciplinary Research and Innovation of Aristotle University of Thessaloniki, 57001 Thessaloniki, Greece; 3School of Dentistry, Aristotle University of Thessaloniki, 54124 Thessaloniki, Greece; tsalikis@dent.auth.gr; 4Laboratory of Pharmaceutical Technology, Division of Pharmaceutical Technology, School of Pharmacy, Faculty of Health Sciences, Aristotle University of Thessaloniki, 54124 Thessaloniki, Greece; pbarmp@pharm.auth.gr; 5School of Chemical Engineering & Physics Laboratory, Faculty of Engineering, Aristotle University of Thessaloniki, 54124 Thessaloniki, Greece; christop@cheng.auth.gr

**Keywords:** periodontitis, 3D-printed scaffolds, tetracycline hydrochloride, MC3T3-E1 pre-osteoblasts, alveolar bone

## Abstract

Current clinical treatment of periodontitis alleviates periodontal symptoms and helps to keep the disease under control for extended periods. Despite this, a significant destruction of the tooth’s underlying bone tissue often takes place progressively. Herein, we present a two-way therapeutic approach for local delivery of antibacterial agents and bone tissue regeneration, incorporating ~1% *w*/*w* tetracycline hydrochloride (TCH) into a 3D-printed scaffold composed of poly(ε-caprolactone) (PCL). Samples were assessed for their morphological, physicochemical, pharmacokinetic, and antibacterial properties. Furthermore, osteoprecursor cells (MC3T3-E1) were employed to evaluate the osteoinductive potential of the drug-loaded scaffolds. Cell proliferation, viability, and differentiation were determined on all cell-seeded scaffolds. At the end of the culture, PCL-TCH scaffolds promoted abundant collagen organic matrix, demonstrating augmented alkaline phosphatase (ALP) activity and areas of accumulated mineralised bone tissue, despite their belayed cell proliferation. Based on the observed effectiveness of the PCL-TCH scaffolds to inhibit *Staphylococcus aureus*, these constructs could serve as an alternative bioactive implant that supports bacterial inhibition and favours a 3D microenvironment for bone tissue regeneration in severe periodontitis.

## 1. Introduction

Periodontitis is a chronic pathological condition that affects the supporting tissues of the periodontium, namely the periodontal ligament, the cementum, and the alveolar bone [1]. This disease is primarily driven by bacterial populations that invade tooth surfaces by creating strongly attached biofilms, which in turn cause inflammation, destruction of the periodontal tissue, and eventually loss of the alveolar bone [2,3]. Common available treatment approaches involve scaling and root planning, which target to mechanically remove dental plaque and periodontal pathogens [4]. In combination with the above treatments, the use of antimicrobial and antiseptic drugs has been reported to act beneficially towards the restoration of the periodontal milieu [3]. However, in many cases, the severity of the disease or the patient’s inadequate response to treatment demand for alternative therapeutic strategies.

Tissue engineering and regenerative medicine have emerged in the field of periodontal tissue regeneration, aiming at restoring oral soft and hard tissues (e.g., alveolar bone) through scaffolds, cells, and signalling growth factors [5]. Specifically, three-dimensional (3D) printing technology has been widely used in biomedical applications, mainly because of its potential to create complex 3D geometric structures of different shape/size with the use of various biomaterials. In this regard, different, yet limited studies have been carried out, aiming to develop 3D-printed scaffolds for regenerating periodontal complex [6]. For example, 3D-printed poly(ε-caprolactone) (PCL)/β-tricalcium phosphate (β-TCP) scaffolds showed superior biocompatibility and bone regeneration compared to 3D-printed PCL samples [7]. Additionally, Lee et al. fabricated 3D-printed PCL/hydroxyapatite scaffolds containing recombinant human amelogenin, connective tissue growth factor, and bone morphogenetic protein-2, and demonstrated that the 3D constructs resulted in dental stem/progenitor cells stimulation and type I collagen synthesis [8]. In a recent study [9], researchers implanted, for the first time, a 3D-printed PCL scaffold containing the recombinant human platelet-derived growth factor BB (rhPDGF-BB) in a patient exhibiting a large periodontal bone defect. It is anticipated that after this first step, similar efforts will be brought to dental clinical practice in the near future. However, the success of any tissue engineering construct relies on its potential to address certain challenges that are evident in periodontal regeneration, such as difficulties in the spatiotemporal compartmentalization of multiple tissues and the orientation of periodontal ligaments [10], among other complexities.

In this vein, a good strategy for treating periodontitis is to employ a drug delivery system that could adequately eliminate both local bacterial infection and facilitate the restoration of the periodontal tissue. This could be achieved using a tissue engineering scaffold loaded with a suitable antimicrobial agent. Various antibiotics have been used in periodontal diseases, with tetracyclines being the most favoured ones [11]. More specifically, tetracycline hydrochloride (TCH), which is effective against a wide spectrum of bacteria, has been employed as the active pharmaceutical ingredient (API) in clinical practice, while in several studies drug-loaded tissue engineering scaffolds have been developed for managing periodontitis [12,13,14,15,16].

Aiming to combine the antibacterial properties of TCH with a 3D printed construct, we developed a TCH-containing (1% *w*/*w* theoretical drug content) 3D-printed PCL scaffold as an alternative two-way therapeutic approach, where the antibiotic release will be used to eliminate periodontitis-associated bacteria, while the scaffold will serve as a 3D matrix for cells to migrate, differentiate, and regenerate bone tissue. PCL was selected as a base material due to its biocompatibility, its relatively slow biodegradability rate, and because it is a non-toxic aliphatic polyester that has been approved by the United States Food and Drug Administration (FDA) for numerous biomedical applications [17]. The fabricated scaffolds were assessed for their physicochemical properties by differential scanning calorimetry (DSC) and thermogravimetric analysis (TGA). Additionally, entrapment efficiency and in vitro drug release characteristics were evaluated. All scaffolds were seeded with MC3T3-E1 pre-osteoblast cells and subsequently characterised for their cell morphology, viability, and attachment. In addition, osteogenic differentiation was assessed in terms of Alkaline Phosphatase (ALP) activity, collagen content (Picrosirius red), and calcium deposition (Calcein staining). Finally, neat and drug-loaded scaffolds were tested for their antibacterial activity against *Staphylococcus aureus*, a strain that is present in periodontal diseases.

## 2. Results

### 2.1. Scaffold Morphological Characteristics

Each specimen was characterised with scanning electron microscopy at an accelerating voltage of 20 kV. Different region of interests (ROIs) were selected from both scaffold types, representative of the geometrical characteristics found on all printed scaffolds. From these ROIs, strand diameters and pore sizes were measured by using the ImageJ software v1.54 (n = 9, per geometrical characteristic). The results are presented in Table 1.

### 2.2. Drug Loaded Scaffolds—Entrapment and In Vitro Release Evaluation

The amount of the incorporated TCH in the PCL scaffolds was assessed spectrophotometrically and expressed as % drug entrapment efficiency (DEE %). PCL-TCH scaffolds demonstrated a very high DEE% of 95% ± 3.6%, and a measured drug loading of 0.95% (±0.04%). With respect to the in vitro dissolution study, the release profile of PCL-TCH scaffolds is shown in Figure 1A. It can be noticed that TCH is released at a faster pace within the first two hours, reaching a cumulative release percentage of ~45%. On the contrary, the next phase (2–72 h) is governed by a controlled drug release, in which approximately a further 30% of the entrapped drug is delivered to the dissolution medium.

### 2.3. Physicochemical Characterisation of Scaffolds

DSC curves of the raw TCH powder along with PCL and PCL-TCH loaded scaffold are presented in Figure 1B. In the case of PCL, the neat scaffold demonstrated a large endothermic peak at 62 °C, which is attributed to PCL’s melting point. In the DSC curve of TCH, the exothermic peak at 228 °C is associated with the onset of thermal decomposition, corresponding to the first mass loss observed in the TGA thermogram (Figure 1C). Regarding the PCL-TCH sample, it can be noticed that PCL’s endothermic peak is present at the same temperature as in PCL scaffold, whereas no peak of TCH could be observed.

TGA analysis for PCL, PCL-TCH scaffold, and TCH powder is shown in Figure 1C. The PCL scaffold begins to decompose at approximately 260 °C, while a second decomposition step can be seen between 400 and 500 °C. The onset decomposition temperature of raw TCH, on the other hand, was recorded at approximately 225 °C. Moreover, two additional mass losses occurred in the TCH sample, one between 240 and 450 °C and another one between 450 and 600 °C. Finally, in the case of PCL-TCH scaffold, the onset of decomposition shifted to a higher temperature, specifically 290 °C.

### 2.4. Cell Growth and Proliferation within Scaffolds

PCL and PCL-TCH scaffolds were evaluated for their biocompatibility and specifically for their cytocompatibility levels at different phases of cell cultivation. CLSM analysis showed that cells adhered and proliferated on both scaffolds. On day 1, cells were more evenly attached and distributed on PCL-TCH scaffolds (Figure 2A VII) compared to PCL scaffolds, in which smaller areas of attached cells can be spotted (Figure 2A I). Despite the considerable number of cells found on PCL-TCH scaffolds, there was also a substantial number of dead cells, almost equal to that of living cells (Figure 2A VII,VIII). The presence of dead cells on PCL-TCH scaffolds was limited on day 7 and day 21, following the same cell viability pattern as in PCL scaffolds (Figure 2A III). Furthermore, on day 21 a more uniform cell colonization was observed on both scaffold variants, covering strand surfaces on top layers and the layers beneath them (Figure 2A XI).

Regarding the evaluation of cell metabolic activity via a resazurin-based assay, cell-seeded scaffolds were measured 5 h after cell seeding and on days 1, 3, 5, and 7 (Figure 2B). PCL and PCL-TCH scaffolds were found to have similar absorbance values 5 h after cell seeding with those measured on day 1, suggesting comparable cell seeding and cell adhesion between scaffolds. Cells that were cultured on PCL scaffolds showed approximately a two-fold increase (1.26 ± 0.05) of their metabolic activity on day 3 compared to the metabolic activity observed on day 1 (0.59 ± 0.08). On day 3, a relatively smaller increase was also evident in the activity of the cells that were cultured on PCL-TCH scaffolds, yet this increase was not statistically significant when compared to the activity of cells on day 1. Cells on PCL scaffolds maintained a significant metabolic activity on day 5 followed by a minor reduction on day 7, while cells’ metabolic activity on PCL-TCH scaffolds was hindered by approximately 57% after day 3 and then remained in the same condition until day 7.

### 2.5. SEM Analysis of Cell-Seeded Scaffolds

Macroscopic characteristics of cells, cellular migration, and newly formed tissue formation on cell seeded scaffolds were assessed with SEM. At day 1, adhered preosteoblasts were observed on the strands’ surfaces of the scaffolds or anchored between the junctions of the strands, as shown in Figure 3 II,VI. Preosteoblasts having elongated shape with extended pseudopodia were evidenced on PCL scaffolds (Figure 3 II), while cells with a more irregular shape were found mostly on PCL-TCH scaffolds (Figure 3 VI). On day 7, cells with extended pseudopodia were spread between scaffold’s layers, displaying a more organized cellular network by highlighting key aspects such as cellular migration and proliferation within PCL scaffolds (Figure 3 III). On the other hand, cells on PCL-TCH scaffolds were detected mainly as single colonies without achieving enough cellular network formation (Figure 3 VII). Preosteoblasts cultured in osteogenic induction medium demonstrated cellular extension with forming sheets of tissue found between the junctions of the scaffolds’ strands for PCL scaffolds (Figure 3 IV), indicative of newly formed ECM. In PCL-TCH scaffolds, clusters of round-shaped cells were found, usually accompanying more elongated cells (Figure 3 VIII). Among these two distinctive cell groups, elongated cells had anchored on the scaffolds’ strands and seemed to be in an arrangement of a more organized forming cellular network, yet without clearly presenting sheets of newly formed tissue as those found on PCL scaffolds.

### 2.6. Antibacterial Performance

The antibacterial activity of the 3D printed scaffolds was assessed against a Gram-positive pathogen, *S. aureus* LMG 8224. Figure 4 demonstrates the antibacterial effect of TCH-containing samples (n = 3) after 24 h of incubation. As it can be observed, the inhibition zones of the drug-loaded scaffolds exhibited an average diameter of 23.04 ± 0.85 mm, confirming the strong antibacterial potential of TCH-loaded scaffolds. On the contrary, drug-free samples showed no inhibition halos around the edges of the scaffolds.

### 2.7. Osteogenic Differentiation and Bone Formation in Scaffolds

#### 2.7.1. Collagen Content

The amount of collagen found on scaffolds was observed macroscopically, under a standard stereoscope (SZ40, Olympus). Collagen was mainly found on the top and bottom layers of the scaffolds, inside their pore network as well as at the circumference of the cylindrical-shaped scaffolds (Figure 5A). In addition, the total amount of collagen per scaffold was measured spectrophotometrically at 535 nm by quantifying the dissolved dye as described in the experimental section, thus avoiding any biased optical observation. The absorbance values of PCL and PCL-TCH samples showed a similar increasing trend from day 7 to day 21. More specifically, both PCL and PCL-TCH scaffolds exhibited approximately 110% increase in collagen content between days 7 and 21. Nevertheless, collagen content was slightly higher (~22%) in PCL scaffolds than in PCL-TCH scaffolds at both timepoints (Figure 5B). This finding is in line with the cell proliferation observed on each scaffold.

#### 2.7.2. Determination of the ALP Activity

Since ALP activity is considered a marker of early osteogenic differentiation [18,19], three timepoints from the 21-day culture period were selected to measure the enzyme’s activity (Figure 6). ALP was mildly expressed on both scaffolds on day 7. Between day 7 and day 14, a two-fold increase was evident only in the PCL scaffolds (*p* = 0.033), while no differences were found in PCL-TCH scaffolds in the same period. Between days 14 and 21, both scaffolds exhibited a steep increase, with PCL following the same motif as in the previous timepoint, whereas PCL-TCH scaffolds showed a steeper increase expressing 126% more of its enzyme’s activity. These findings strongly support the idea that pre-mature osteoblasts had differentiated and led to an early osteogenesis, even if PCL-TCH scaffolds presented a lag phase to osteoblast maturation.

#### 2.7.3. Bone Tissue Formation and Labelling

No fluorescence signal was detected on PCL-TCH scaffolds on day 7, while minor signs of green stain dots were observed on PCL scaffolds. On day 14, PCL scaffolds demonstrated areas of mineralisation, mainly at the edges of the strands and close to the pores of the scaffold (Figure 7 I). On the same day, scattered green stained dots were observed on the strands of PCL-TCH scaffolds (Figure 7 III) indicating the initiation phase of mineral deposition. On day 21, larger areas of intense, green-stained strands were evident on PCL scaffolds (Figure 7 II), while more fuzzy green areas were observed on PCL-TCH scaffolds (Figure 7 IV). These findings indicate that both scaffolds were found with areas covered of mature bone tissue formation, still with recognizable differences between them.

## 3. Discussion

Clinically, the first stages of periodontitis treatment include the removal of the dental plaque and the calculus (gingivitis), administration of antibiotics, minimally invasive nonsurgical therapy (MINST) of the periodontal pockets, and laser-assisted periodontal therapy for cases of moderate periodontitis [20,21,22,23,24]. Although these treatments are very important, in many cases a secondary infection can be developed leading to partial or complete bone loss. Current treatments for aggressive periodontitis include the access surgery (modified Widman flap procedure), as well as the employment of different regenerative techniques, such as (i) bone grafting (autografts, allografts), (ii) guided tissue regeneration (GTR) using non-resorbable and resorbable membranes, and (iii) biological modifiers (enamel matrix proteins) [25,26]. Nevertheless, the above treatments may present certain drawbacks, such as the limited amount of graft material, donor-site morbidity, additional surgical procedures, and graft rejection by the recipient’s immune system, among others [27]. Tissue engineering can offer an alternative method for the complete treatment of periodontitis and regeneration of the lost bone tissue. This approach utilises the use of 3D bioactive scaffolds that mimic the basic biological microenvironment for cells to proliferate, differentiate, and finally to regenerate the targeted tissue, while the same time can be also used as drug delivery systems.

This work focuses on the fabrication of 3D printed PCL scaffolds incorporating TCH that can constitute an alternative bioactive implant, providing local antimicrobial protection against periodontitis-associated bacteria and a favourable 3D micro-environment for osteoblast migration, differentiation, and bone regeneration. To the best of our knowledge, it is the first time that such scaffolds are produced for the treatment of severe periodontitis.

### 3.1. Bioactive Scaffolds and Their Physiochemical Characteristics

All our fabricated 3D-printed scaffolds (neat and drug-loaded) exhibited satisfactory geometrical characteristics, with pore sizes of ~360 μm (within the range of 150 to 600 μm), which favours the ingrowth of vascular tissues in bone regeneration applications [28,29,30]. Furthermore, the preparation of the PCL-TCH material resulted in printed scaffolds with exceptionally high DEE. Analogous encapsulation efficiency percentages were also published by other research groups regarding different delivery platforms for TCH, such as electrospinning and solvent casting techniques [16,31]. With respect to the in vitro dissolution study, our findings demonstrated that TCH is released from the 3D scaffolds at a faster initial pace, followed by a more sustained release rate during the 2–72 h window. This observation is possibly associated with the hydrophilic nature of the drug in combination with the surface-bound drug molecules that get released easier [16,31,32]. Initial burst drug release is a phenomenon that is witnessed in various drug delivery platforms, and it can be controlled by modulating specific parameters, such as the surface of the drug-loaded matrix, its geometry and structure, as well as the type of materials used [33].

Thermal analysis of the drug-loaded scaffolds indicated that the large exothermic peak of TCH, which corresponds to the onset of thermal decomposition of the drug, could not be observed in the PCL-TCH thermogram. This can be probably explained by the relatively small amount of drug quantity (nominally 1% *w*/*w*) present in the PCL-TCH scaffolds. Another interesting observation is the absence of a distinct melting peak (endothermic) of TCH. Despite literature data supporting that the melting of TCH occurs at 220–223 °C [34], DSC analysis revealed a minuscule endothermic signal at ~217.5 °C, which can be attributed to TCH’s melting. However, this thermal event is accompanied by the onset of thermal decomposition of TCH (exothermic peak at 228 °C), as it has been documented by other researchers [35].

### 3.2. Bioactive Scaffolds and Their Potential to Bone Regeneration

Following the physicochemical characterisation, a biological evaluation with osteoprecursor cells was conducted. Murine pre-osteoblasts (MC3T3-E1) displayed a satisfactory cell attachment on both scaffold types, with a more uniform cell distribution on PCL-TCH scaffolds than on PCL scaffolds. However, on day 1, PCL-TCH scaffolds exhibited a considerable number of dead cells compared to live cells. Khodir et al. (2013) reported improved cell viability at day 1 when mesenchymal stem cells had a direct contact with particles containing tetracycline–chitosan (1% w/w) that were spread over PCL electrospun scaffolds [16]. This is likely due to the different drug release kinetics and the drug entrapment methodology on their scaffolds compared to ours and the different cell line employed. During the next days of culture, cells remained metabolically active and proliferated on PCL scaffolds as expected and consistently with similar scaffolds [36,37,38]. Cell proliferation was inhibited by PCL-TCH scaffolds from day 3 to day 7 of culture; this agrees with the study of Ferreira et al. that combined TCH with polydioxanone (PDS) [39]. However, it is notable that PCL-TCH scaffolds exhibited an insignificant number of dead cells on day 7, comparable to those found in PCL scaffolds, as observed by the CLSM analysis in Figure 2A. Therefore, it appears that tetracycline antibiotics act as a barrier, hindering cell proliferation but without altering their viability on prolonged cultures. This can probably be explained by the fact that tetracyclines binds to cellular RNAs, and, as a consequence, this may alter the normal functioning of different biological processes that the RNAs regulate [40,41]. Park et al. reported that cell viability resulted in insignificant differences when preosteoblasts were cultured without or with tetracycline, in the range of 0.1 and 1.0 μM [42]. This is in agreement with our viability results observed on day 7. Despite this, the same researcher, in another study, exclaimed that higher doses of tetracycline in the range of 100–1000 μM can yield a negative effect on cell viability [43].

Regarding cell differentiation, bone formation, and mineralisation on 3D scaffolds loaded with tetracycline, very little data exist in the literature, supporting that tetracyclines at low concentrations may facilitate osteogenesis [44]. Nevertheless, several tentative mechanisms have been proposed through which tetracycline antibiotics modulate osteogenic differentiation. For example, tetracycline hydrochloride (at 1 μg/mL) was found to promote in vitro osteogenesis in human bone marrow stem cells by upregulating the Wnt signalling pathway [45]. Also, this class of compounds seems to inhibit the activity of various matrix metalloproteinases (MMPs), therefore impeding osteoclast activity that leads to osteopenia and osteoporosis [44,46,47,48,49]. In our study, osteoprecursor cells were used as a model to evaluate osteoblast differentiation, as it is well known that these cells can differentiate to more mature osteoblasts leading to bone tissue formation and mineralisation [50,51]. Collagen, one of the most abundant proteins found in bones (90% of the organic matrix) [52], was found on both PCL and PCL-TCH scaffolds at the end of the culture, confirming the production of the basic organic structural matrix. ALP enzyme activity as an indicator of the early phase of osteoblast maturation was upregulated on both scaffolds in a time-dependent manner. Specifically, in PCL-TCH scaffolds, ALP was upregulated on day 21, suggesting that PCL-TCH scaffolds promoted osteogenic differentiation, probably when a greater amount of the drug has been released from the scaffold, as shown from the in vitro dissolution study. Similar results have been reported in another study with 3D-printed composite scaffolds (β-TCP/PLGA-PCL) containing tetracycline hydrochloride after 14 days of culture [32]. Similar results were presented in a study by Dayaghi et al., in which magnesium–zinc scaffolds that had been immersed in solutions of different tetracycline concentrations (10–100% *w*/*v*), showed that ALP levels were increased in all scaffolds’ concentrations with increasing incubation time [53]. As the enzyme activity is considered a major regulator for osteoblast differentiation and the initiation of mineralisation and hydroxyapatite formation [19], our findings further supported mineralised surfaces on both PCL and PCL-TCH scaffolds. However, freeze dried PCL-PLA-TCH scaffolds had better supported bone healing effects than that of drug-free scaffolds when assessed eight weeks after in an in vivo study with Wistar rats [54]. With respect to the varying designed parameters of each experimental study, i.e., in vitro or in vivo studies, TCH-loaded scaffolds do not seem to have a negative effect on bone re-generation or bone healing applications.

### 3.3. Challenges and Future Perspectives

We present a two-way therapeutic approach for the local delivery of the antibiotic agent tetracycline hydrochloride. Our bioactive construct exhibited an enhanced antibacterial activity, while at the same time it remained highly biocompatible, without showing any cell-toxic effects, during prolonged cell culture. Despite the initial delay in cell proliferation, as witnessed by cell viability studies, in vitro osteogenesis was successful after 21 days in the presence of TCH-loaded samples. Thus, it can be concluded that our drug-loaded scaffolds may serve as an alternative bioactive implant for regenerating bone tissue in severe periodontitis. This research brings promising results in the field of applying tissue engineering API-loaded scaffolds to periodontal diseases.

Further steps would be interesting in order to fully explore the scaffolds’ potential and shed light on the cellular and molecular mechanisms that govern osteogenesis, in the presence of TCH-containing scaffolds. Additionally, the concentration of TCH within scaffolds should be optimized in terms of cell proliferation and differentiation and the interplay between drug and bone regeneration should be also assessed in vivo in long-term studies. This will allow us to evaluate the safety and the efficacy of TCH-loaded scaffolds, which is an important preclinical step before transitioning to clinical trials.

We anticipate that our drug-containing engineered scaffold could be considered as a suggestion for other polymer-based 3D scaffolds that have shown to enhance bone regeneration. For example, polymers that have been combined with well-known osteo-inductive materials, such as hydroxyapatite or β-TCP [55,56,57], could serve as platforms for antibiotic and/or anti-inflammatory compounds for bone regeneration applications [58,59]. Such a strategy, to incorporate both anti-inflammatory and antibiotic APIs in 3D matrices, could be relevant in periodontitis, where bacteria and inflammation co-exist, comprising an alternative solution to the current periodontal therapies. Hence, it will be interesting to study the synergistic effect of both anti-inflammatory and antibiotic drugs on bone remodelling and bone regeneration and investigate their release kinetics from such scaffold matrices.

## 4. Materials and Methods

### 4.1. Material Preparation, for Neat and Drug-Loaded Scaffolds

Poly(ε-caprolactone) (PCL) 3D filament with an average molecular weight of 50 kDa and a density of 1.145 g/cm^3^ (3D4MAKERS, Haarlem, Netherlands) was used as a base material for the fabrication of neat scaffolds. For the drug loaded scaffolds, the PCL filament was further chopped with a scissor and 20 g of pellets were placed in oven and heated at 140 °C for 30 min to allow the polymer to melt. Simultaneously, 0.22 g of tetracycline hydrochloride powder (Sigma-Aldrich, St. Louis, MO, USA) was added into the melted PCL and mixed thoroughly for 3 min to yield a composite material with 1.088% (*w*/*w*) of theoretical drug content (~1% *w*/*w*). The blended material was placed in a petri dish and left for 40 min at room temperature (RT) to solidify and become a thin film. The film was further cut into pellets to fit in the cartridge of a BIO-X 3D printer (Cellink, Gothenburg, Sweden).

For the fabrication of neat and drug-loaded scaffolds, a rectilinear shape (21.00 × 21.00 × 3.42 mm) was employed, and the internal pattern geometry was sliced with Slice3r found in the Cellink HeartWare (Cellink, Gothenburg, Sweden) software version 2.4.1. The pattern was composed of 9 layers, 380 μm thickness each, and strands of 400 μm in diameter. The layers had a rectilinear relationship to each other (0°–90°–0°), resulting in rectilinear pores of 325 μm. The printing temperature was set at 100 °C, the extrusion speed at 17 mm/s, and the air pressure of the pneumatic thermoplastic printhead to 340 kPa. For the experiment, a biopsy punch (GIMA, Milan, Italy) of 6 mm in diameter was used to cut cylindrical scaffolds from the fabricated rectilinear meshes of both neat PCL and drug-loaded PCL scaffolds, designated as PCL and PCL-TCH, respectively. Prior to cell culture experiments, all scaffolds were sterilized on both surfaces for 1h per surface under UV light inside a laminar flow hood (Safemate 1.2, Class II Microbiological Safety Cabinet, BioAir, Pavia, Italy).

### 4.2. Drug Entrapment Efficiency

The drug entrapment efficiency (DEE %) was determined by quantifying the incorporated drug compound detected within scaffolds. For this test, pre-weighted samples of the drug-loaded scaffolds (~15 mg per sample, n = 3) were immersed in a 50 mL falcon tube containing 10 mL of chloroform and vortexed to fully dissolve the scaffold structure. Next, 10 mL of phosphate-buffered saline (PBS) were added to the solution to extract the drug. The aqueous phase containing the dissolved drug was collected, filtered, and measured by UV/Vis spectrophotometry (UV-1900 spectrophotometer, Hitachi, Tokyo, Japan) at λ_max_  =  363 nm. The drug entrapment efficiency was calculated based on Equation (1):(1)$$\mathrm{DEE}\;\%=\left(\frac{\mathrm{amount}\;\mathrm{of}\;\mathrm{drug}\;\mathrm{entrapped}\;\mathrm{in}\;\mathrm{scaffolds}}{\mathrm{amount}\;\mathrm{of}\;\mathrm{drug}\;\mathrm{added}\;\mathrm{initially}}\right)\times 100\;$$

A calibration curve was constructed for calculating drug content based on absorbance values. For this curve, different TCH concentrations were dissolved in PBS (n = 8) and the results were plotted against absorbance values obtained spectrophotometrically.

### 4.3. In Vitro Release Kinetics

A solution containing phosphate-buffered saline (PBS)/sodium dodecyl sulfate (SDS) 1% *v*/*v* was stirred at low rpm to obtain a homogeneous mixture. Samples of the drug-loaded scaffolds (~110 mg per sample, n = 3) were immersed in 30 mL of the dissolution medium and incubated at 37 °C (mod. MIR-153 incubator, SANYO Electric Co., Ltd., Osaka, Japan) under moderate shaking. At specific timepoints, 3 mL aliquots of the release medium were collected, and an equal amount of fresh medium was added each time to maintain sink conditions. Released TCH in the dissolution medium was estimated using a UV/Vis spectrophotometer at 363 nm through the calibration curve, based on Equation (2):(2)$$\mathrm{Drug}\;\mathrm{concentration}\;\left(\frac{\mathrm{mg}}{\mathrm{mL}}\right)=0.035\times \;\mathrm{absorbance}\;+0.0025\;;\;\left(R^{2}=0.998\right)$$

The cumulative TCH release was calculated and plotted as a function of time, based on Equation (3):(3)$$\mathrm{Cumulative}\;\mathrm{drug}\;\mathrm{release}\;\%=\left(\frac{\mathrm{drug}\;\mathrm{released}}{\mathrm{entrapped}\;\mathrm{drug}}\right)\times 100$$

### 4.4. Scaffold Characterisation

#### 4.4.1. Morphological Characterisation

PCL and PCL-TCH scaffolds were observed using a scanning electron microscope (JSM-6390LV, JEOL, Tokyo, Japan). Briefly, scaffolds were sputter coated with carbon, and visualized at an accelerating voltage of 20 kV. Strands and pores were selected from different region of interests (ROIs) of both scaffold types, and their sizes measured by using the ImageJ software v1.54 (National Institutes of Health, Bethesda, MD, USA).

#### 4.4.2. Thermal Analysis

Differential scanning calorimetry measurements were carried out under a nitrogen atmosphere (constant flow of 20 cm^3^ min^−1^) on a Shimadzu differential scanning calorimeter (DSC-50, Shimadzu, Kyoto, Japan). Samples with a weight of <10 mg were heated to 250 °C with a heating rate of 10 °C min^−1^. Thermal gravimetric analysis was also performed under a constant nitrogen flow (20 cm^3^ min^−1^) on a Shimadzu TGA-50 thermogravimetric analyser. Samples were heated up to 600 °C with a heating rate of 10 °C min^−1^.

### 4.5. Cell Seeding, Proliferation and Osteogenic Induction

All cell culture experiments have been approved by the Ethics and Research Integrity Committee of the Aristotle University of Thessaloniki, Greece (#331195/2022). Murine pre-osteoblast cells MC3T3-E1 (American Type Culture Collection, Manassas, VA, USA) were cultured in expansion complete medium containing a-MEM (Minimum Essential Medium Eagle Alpha, Sigma-Aldrich, St. Louis, MO, USA), 10% FBS (Gibco™, Thermo Fisher Scientific, Waltham, MA, USA), and 1% antimicrobial solution (Penicillin-Streptomycin-Amphotericin B, PAN-Biotech GmbH, Aidenbach, Germany). MC3T3-E1 cells were expanded until passage 4 and counted using a Neubauer chamber. Scaffolds were placed to 24-wells and 30 μL of culture medium containing 0.25 × 10^6^ cells were suspended on each scaffold. After suspending cells onto the scaffolds, cell-seeded scaffolds were maintained inside the incubator (37 °C, 5% CO_2_ and 21% O_2_) for at least 1h in order to allow cell attachment, before 1 mL of expansion medium was added. After 7 days, the expansion medium was replaced with osteogenic medium, based on complete culture medium, which was further supplemented with 100 nM dexamethasone (Merck KGaA, Darmstadt, Germany), 50 μg/mL L-ascorbic acid 2-phosphate, and 10 mM of β-glycerophosphate (Cayman Chemical Co., Ann Arbor, MI, USA). Scaffolds were cultured for 21 days, and medium changes were performed every 2–3 days, as needed.

### 4.6. Cell Viability Assessment

#### 4.6.1. Cell Viability Assessment with the AlamarBlue™

Viable and metabolically active cells within scaffolds were measured by using the resazurin based assay, alamarBlue™ (Biotium, Fremont, CA, USA). Resazurin, as a non-toxic compound, penetrates active cells where it reduces to resorufin. During this reduction, the colour of resazurin (purple, non-fluorescent) changes to a fluorescent red colour (resorufin) and thus the overall fluorescence of the cell culture media is increased. Cell viability was measured at four different timepoints at days 1, 3, 5, and 7. Each scaffold variant (n = 5) was treated with fresh media containing 10% (*v*/*v*) of alamarBlue™ and incubated for 4 h, before collecting the coloured media and replacing them with fresh media. The coloured media were measured spectrophotometrically under UV/Vis spectrophotometry (UV-1900 spectrophotometer, Hitachi, Tokyo, Japan) at 570 and 600 nm.

#### 4.6.2. Cell Viability Assessment by Live/Dead Staining

Live and dead cells on all scaffold variants were stained with a viability/cytotoxicity assay kit (Biotium, Fremont, CA, USA) at days 1, 7, and 21 and assessed with an inverted confocal laser scanning microscope (CLSM, Zeiss, Axio Observer.Z1, Zeiss, Berlin, Germany) equipped with the LSM 780 laser scanning unit. The CLSM facility was located at the Department of Botany, School of Biology, Aristotle University of Thessaloniki. For the excitation of MC3T3-E1 cells, the 488 nm (for Calcein-AM) and 514 nm (for EthD-III) lasers were used, while the detection of the emitted light was performed at 543 and 633 nm, respectively. Briefly, cell-seeded scaffolds were collected from the culture plates, rinsed with PBS, immersed in the solution containing both calcein AM (living cells—green colour)/ethidium homodimer III (dead cells—red colour), and remained in the dark for at least 45 min. Prior to imaging, scaffolds were washed with PBS three times to clear the residual dye. Digital images were acquired using the ZEN software v.2011 (Carl Zeiss, Berlin, Germany).

### 4.7. Cell-Seeded Scaffolds Morphology

Morphology of cells and the newly formed extracellular matrix was observed at different timepoints visualizing the sample surfaces by means of SEM. Scaffolds were fixed with 4% *w*/*v* formaldehyde, rinsed in PBS, and dehydrated in increasing ethanol series (30–100% *v*/*v*). The samples were air dried, sputter coated with carbon, and observed under SEM (JEOL JSM-6390 LV, Tokyo, Japan) at an accelerating voltage of 20 kV.

### 4.8. Preosteoblasts Differentiation and Osteogenesis

#### 4.8.1. Alkaline Phosphatase Activity

The alkaline phosphatase (ALP) assay kit (Sigma Aldrich, St. Louis, MO, USA) was used to detect the enzyme’s activity on cell lysates of all scaffold variants at days 7, 14, and 21. To obtain cell lysates from scaffolds, five samples (n = 5) per scaffold variant were washed with PBS twice and then immersed in an Eppendorf tube containing a solution of PBS with 0.2% v/v of Triton™ X-100 (Sigma Aldrich, St. Louis, MO, USA). To further support cell lysis, each sample underwent an ultrasonication treatment (Sonorex Digital 10P, Bandelin, Germany) for 10 min. Prior to testing, all Eppendorfs containing cell lysates were centrifuged (Z216MK, HERMLE, Reichenbach am Heuberg, Germany) at 18,000× *g* for 10 min at 4 °C. From each Eppendorf, 100 μL of suspension was mixed with 100 μL of ALP working reagent (containing 1M of p-nitro-phenyl-phosphate) and incubated at 37 °C for 45 min. The intracellular ALP activity was measured spectrophotometrically at 405 nm (UV-1900 spectrophotometer, Hitachi, Tokyo, Japan). Data were normalized to the total cell protein measured with the BCA Protein Assay Reagent Kit (AppliChem GmbH, Darmstadt, Germany). Thus, the ALP activity was expressed as μmol L^−1^ min^−1^ per μg of total protein.

#### 4.8.2. Collagen Content

The total amount of collagen found on scaffolds was evaluated for at days 7 (proliferation phase) and 21 (differentiation phase) using picrosirius red dye (Sigma-Aldrich, St. Louis, MO, USA). Scaffolds were fixed in 4% *w*/*v* formaldehyde, washed with PBS, and immersed in picrosirius red for 24 h. Scaffolds were then washed multiple times with distilled water to remove any unbound dye and allowed to air dry. The total amount of collagen per scaffold was quantified by dissolving the bounded dye. Briefly, scaffolds were immersed in a solution containing PBS and 0.1 M NaOH and incubated at 37 °C, under moderate shaking at 30 rpm, for at least 30 min. 300 μL of the dissolved medium was collected from each sample and measured spectrophotometrically at 535 nm (UV-1900 spectrophotometer, Hitachi, Tokyo, Japan).

#### 4.8.3. Mineralised Bone Tissue Labeling

Calcein green fluorochrome as a calcium chelator has extensively been used in in vivo and ex vivo studies [60,61,62], mainly to evaluate bone metabolism, bone growth/remodeling, and the localization of new bone to regions of mineralising bone. Thereby, calcium-dependent fluorescent molecule (Calcein, Sigma-Aldrich, St. Louis, MO, USA) was adopted in our study to label newly formed bone tissue and mineralised regions of bone matrix. To assess mineralising bone surfaces, calcein fluorochrome was added to cell culture media on day 7, 14, and 21 at a concentration of 50 μg/mL. The media containing calcein fluorochrome were replaced with fresh media after 24 h. At the end of the experiment, stained scaffolds were rinsed with PBS, fixed with 4% *w*/*v* formaldehyde, and observed under an inverted confocal laser scanning microscope (Zeiss, Axio Observer.Z1, Zeiss, Berlin, Germany) equipped with the LSM 780laser scanning unit. Fluorometric analysis was performed using the ZEN software v.2011 (Carl Zeiss, Berlin, Germany).

### 4.9. Antibacterial Assessment

The antibacterial activity of TCH-loaded PCL scaffolds was tested against Gram-positive *Staphylococcus aureus* LMG 8224 (Belgian Co-ordinated Collections of Micro-organisms, Belgium), as previously described [63,64]. Briefly, *S. aureus* was first cultured on nutrient agar (Merck, Germany) plates (24 h at 37 °C) before being transferred to Muller–Hinton agar (Merck, Germany) plates for conducting the test. UV-sterilized PCL-TCH scaffolds with an average diameter of 6 mm were placed on the surface of inoculated MH plates and incubated at 37 °C for 24 h. At the end of the incubation period, the antibacterial activity was assessed by measuring the inhibition zone of *S. aureus* in millimetres.

### 4.10. Statistical Analysis

Statistical analysis for cell viability and metabolic activity was performed using one-way ANOVA with Dunnett’s multi-comparison test comparing the different experimental timepoints in PCL-TCH and control PCL samples. For ALP activity, the one-way ANOVA model was used followed by Tukey’s post-hoc test at each experimental time point. For collagen content quantification, a two-way ANOVA was used at each experimental time point and for each group. All tests were performed in GraphPad Prism version 9 software (GraphPad Software, San Diego, CA, USA). Differences between mean values were considered statistically significant when * *p* < 0.05, ** *p* < 0.01, *** *p* < 0.001, **** *p* < 0.0001. Data were presented as mean ± standard deviation (SD) for n = 5 samples per test.

## Figures and Tables

**Figure 1 ijms-24-16754-f001:**
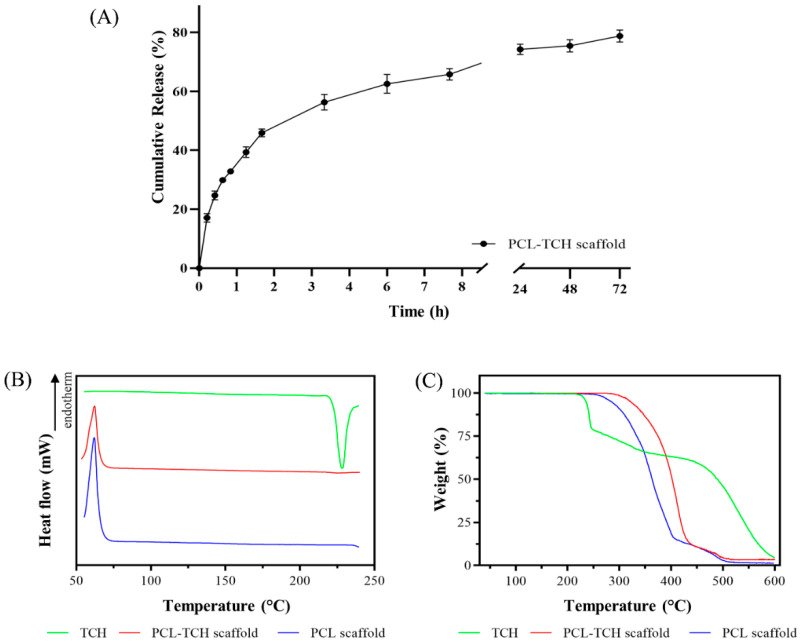
(**A**) Cumulative drug release profile of the entrapped TCH versus time from PCL-TCH scaffolds. Each value represents the mean ± SD of three independent experiments (n = 3). (**Β**) DSC analysis curves of neat and TCH-loaded PCL scaffolds. (**C**) TGA thermographs for neat and TCH-loaded PCL scaffolds.

**Figure 2 ijms-24-16754-f002:**
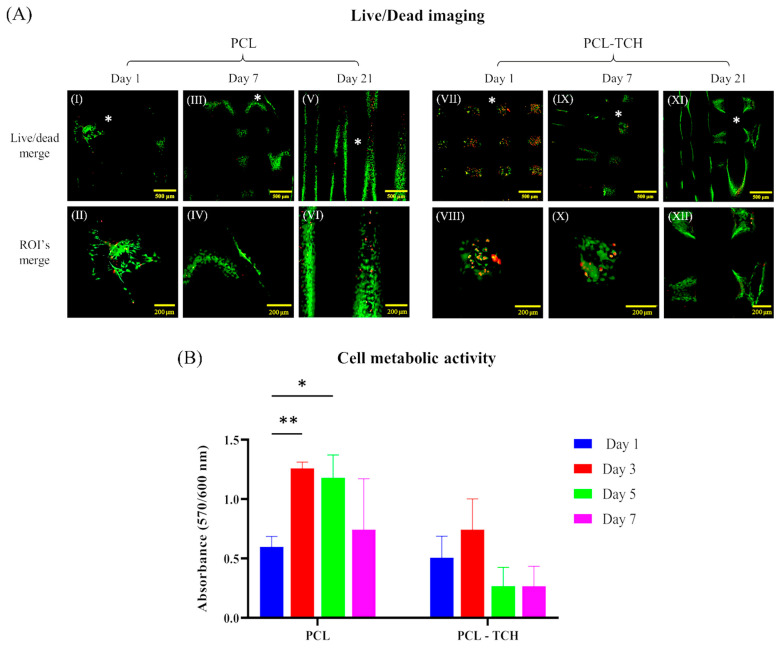
(**A**) Cell viability assessment with Confocal Laser Scanning Microscopy (CLSM). Representative CLSM (live and dead cells) images on day 1, 7 and 21. PCL scaffolds for days 1, 7 and 21 are illustrated in (**I**), (**III**) and (**V**), respectively. PCL-TCH scaffolds for days 1, 7 and 21 are illustrated in (**VII**), (**IX**) and (**XI**), respectively. High magnification images of represented Regions of Interest (ROI’s) are shown with asterisks for all scaffolds and timepoints. ROIs of PCL scaffolds are illustrated in (**II**), (**IV**), and (**VI**) for days 1, 7 and 21, respectively. ROIs of PCL-TCH scaffolds are illustrated in (**VIII**), (**X**), and (**XII**) for days 1, 7 and 21, respectively. (**B**) Cell metabolic activity with alamarBlue™ assay. Assessment of MC3T3-E1 pre-osteoblast cells cultured on PCL and PCL-TCH scaffolds and measured on days 1, 3, 5 and 7. Each bar represents the mean ± SD of n = 5 samples (* *p* < 0.05, ** *p* < 0.01), comparing the different experimental timepoints against day 1 of each sample group. Statistical analysis was performed using one-way ANOVA with Dunnett’s multi-comparison test.

**Figure 3 ijms-24-16754-f003:**
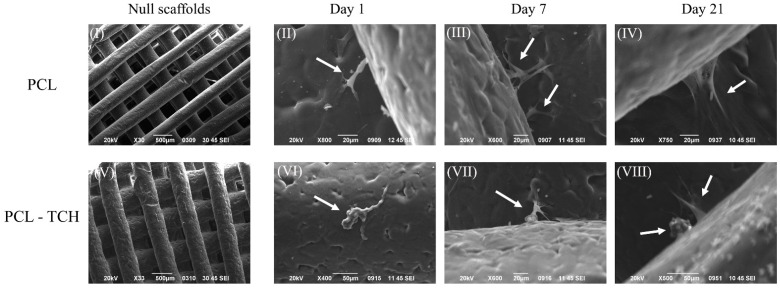
Scanning electron microscopy (SEM) images of null scaffolds (prior to cell seeding): PCL null scaffold (**I**) and PCL-TCH null scaffold (**V**). Cell-seeded scaffolds were cultured on complete culture medium for 7 days. PCL scaffolds cultured for days 1 and 7 are illustrated in (**II**) and (**III**), respectively. PCL-TCH scaffolds cultured for days 1 and 7 are illustrated in (**VI**) and (**VII**), respectively. PCL (**IV**)and PCL-TCH (**VIII**) on day 21 illustrate cell growth and differentiation when cultured under osteogenic medium. On day 1, arrows point out adherent cells and their morphologies on both scaffold variants. On day 7, cells with extended pseudopodia were found between scaffold’s layers and on day 21 forming sheets of tissue were evident and are highlighted, mainly on PCL scaffolds.

**Figure 4 ijms-24-16754-f004:**
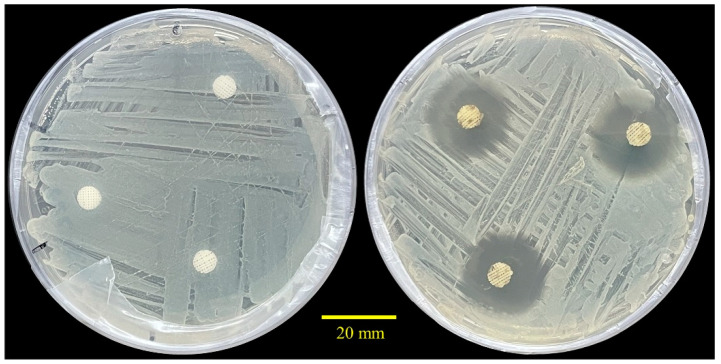
Zone of inhibition against *S. aureus* LMG 8224, after 24 h of incubation: PCL (**left**) showed no inhibition, and PCL-TCH (**right**) had an inhibition of 23.04 ± 0.85 mm.

**Figure 5 ijms-24-16754-f005:**
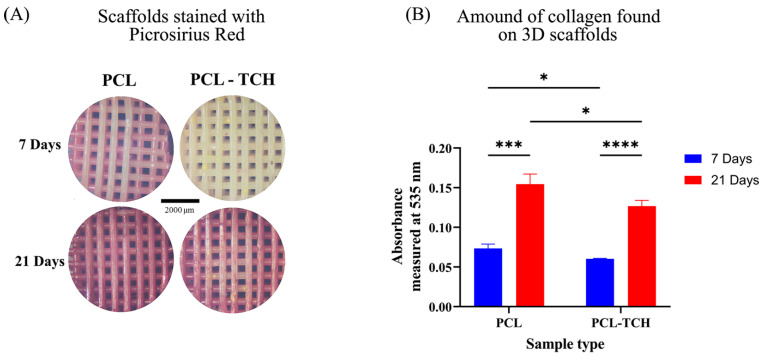
(**A**) Macroscopic images of scaffolds stained with picrosirius red on day 7, after cell proliferation with basic culture medium and on day 21 after cell differentiation with osteogenic medium. (**B**) The corresponding collagen content, quantified by dissolving the red dye from the stained scaffolds and measuring its absorbance spectrophotometrically at 535 nm. Each bar represents the mean ± SD of n = 5 samples (* *p* < 0.05, *** *p* < 0.001, **** *p* < 0.0001). The results were analysed using two-way ANOVA at each experimental time point and for each group.

**Figure 6 ijms-24-16754-f006:**
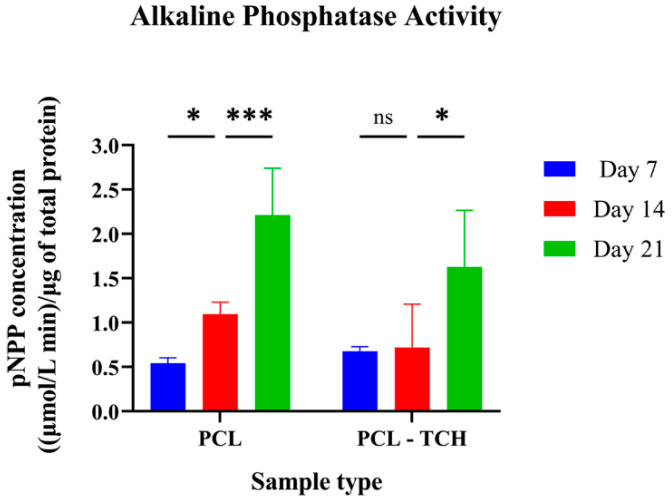
(**A**) Normalized alkaline phosphatase activity of MC3T3-E1 cells in PCL and PCL-TCH scaffolds on days 7, 14, and 21. Each bar represents the mean ± SD of n = 5 samples. (* *p* < 0.05, *** *p* < 0.001, ns: non-significant). For ALP activity, results were analysed with one-way ANOVA followed by Tukey’s post-hoc test comparing the different timepoints against day 7 for each sample group.

**Figure 7 ijms-24-16754-f007:**
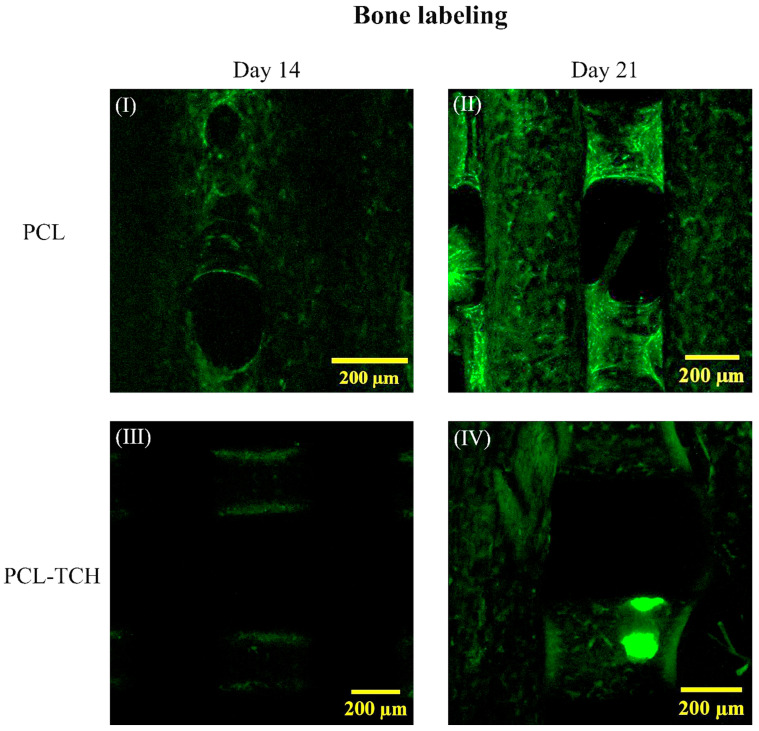
Representative fluorescent images of both scaffold variants stained with Calcein as an indicator of bone labelling on days 14 and 21 of culture. On day 14, magnified images demonstrate bone mineralisation at the edges of several strands of the PCL scaffold (**I**). Magnified images on day 21 revealed larger areas of mineralised bone tissue surrounding the strands of the PCL scaffolds and to the junctions between the scaffold’s layers (**II**). Images from PCL-TCH scaffold on day 14, barely show traces of mineralisation that were found mainly on single strands (**III**), PCL-TCH scaffolds showed signs of initial mineralisation and bone formation at the end of the culture (**IV**).

**Table 1 ijms-24-16754-t001:** Measurements of strand diameters and pore sizes of the 3D-printed scaffolds.

Scaffold Theoretical and Measured Characteristics				
Scaffold Type	Theoretical Strand Diameter (μm)	Measured Strand Diameter (μm)	Theoretical Pores Sizes (μm)	Measured Pores Sizes (μm)
PCL	400	358.26 ± 24.09	325	355.32 ± 12.33
PCL-TCH	400	360.59 ± 17.09	325	364.17 ± 25.18

n = 9 per geometrical characteristic.

## Data Availability

Data is contained within the article.

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
