# Peer review of "3D-Printed Antibacterial Scaffolds for the Regeneration of Alveolar Bone in Severe Periodontitis"

_ijms, 2023, doi:10.3390/ijms242316754_

Round 1

Reviewer 1 Report

Comments and Suggestions for Authors

The manuscript presents the 3D printing of a PCL/TCH scaffold and the use of preosteoblasts (MC3T2-E1) to study the effects on bone differentiation properties and the characterization of antibacterials. The presentation of a two-way therapeutic approach for bone tissue regeneration and local delivery of antibacterials using a biocomposite (PCL/TCH) is appropriate, but errors exist in the manuscript's clarity and experiments. Major revision is required before acceptance.

1.       The introduction section should be strengthened to include a various research examples of prior research related to 3D bioprinting for alveolar bone regeneration.

2.       Why was the drug concentration (~1% w/w of TCH) fixed to one condition? The number of experimental groups is not enough to justify the optimal amount of drug. Additionally, How much drug was in the final fabricated scaffold?

3.       Tetracycline hydrochloride is commonly known to be thermally degraded and has been shown to be unstable (Ref: Y. Wu and R. Fassihi, Stability of metronidazole, tetracycline HCl and famotidine alone and in combination, International Journal of Pharmaceutics, 2005, 290, 1-2, 1-13.) However, the authors processed it at temperatures as high as 140oC to fabricate the composite scaffold. Please provide evidence for the stability of the drug in this regard.

4.     The authors stated that there was no direct prior research showing that osteoblast differentiation was activated by tetracycline hydrochloride. To reinforce this, please add information about prior research to the discussion section. Or, strengthening data is needed to prove it experimentally. (Example: qRT-PCR data using various specific genes related to osteogenesis that can confirm osteogenic diffrentiation, etc.)

References

-       S. Farzamfar et al., Tetracycline hydrochloride-containing poly (e-caprolactone)/poly lactic acidscaffold for bone tissue engineering application:in vitroandin vivostudy, Tetracycline, International Journal of Polymeric Materials and Polymeric Biomaterials, 2019, 68, 8, 472–479

-       E. Dayaghi et al., Magnesium-zinc scaffold loaded with tetracycline for tissue engineering application: In vitro cell biology and antibacterial activity assessment, Materials Science and Engineering: C, 2019, 102, 53-65.

-       W. Wang et al., Effect of tetracycline hydrochloride application on dental pulp stem cell metabolism–booster or obstacle for tissue engineering?, Frontiers in Pharmacology, 2023, 14, 1277075.

5.       Surface and cross-sectional SEM images of the control group (PCL) and experimental group (PCL/TCH) should be added to the manuscript.

6.       The resolution of Figure 2A and Figure 7 images must be improved as well as the high magnification images must be provided.

Author Response

We would like to thank the Reviewer for the time devoted to review our manuscript and for hir/her appreciation of our work, the valuable comments, as well as constructive criticism. Following his/her valuable suggestions, the manuscript has been improved and all the points raised are answered one by one.

Reviewer 1

The manuscript presents the 3D printing of a PCL/TCH scaffold and the use of preosteoblasts (MC3T2-E1) to study the effects on bone differentiation properties and the characterization of antibacterials. The presentation of a two-way therapeutic approach for bone tissue regeneration and local delivery of antibacterials using a biocomposite (PCL/TCH) is appropriate, but errors exist in the manuscript's clarity and experiments. Major revision is required before acceptance.

  1. The introduction section should be strengthened to include a various research examples of prior research related to 3D bioprinting for alveolar bone regeneration.

As kindly indicated by the reviewer, we tried to enhance the introduction part by citing additional literature data that demonstrate the utilization of 3D-printed scaffolds in applications targeting alveolar bone regeneration.

  1. Why was the drug concentration (~1% w/w of TCH) fixed to one condition? The number of experimental groups is not enough to justify the optimal amount of drug. Additionally, How much drug was in the final fabricated scaffold?

As stated in the introductory section, we developed TCH-containing (1% w/w theoretical drug content) 3D-printed PCL scaffolds. So, the experimental design was performed for the 1% drug concentration. This concentration was selected based on discussions with our team of Clinical Dentists (Prof Tsalikis, co-author), who are using in the clinical practice mainly gels with 1% concentration of antibiotics, including TCH. We did not state that we found the optimal amount of TCH, since we only tested the concentration of 1% w/w. Furthermore, in the discussion part we state that “the concentration of TCH within scaffolds should be further optimized in terms of cell proliferation and differentiation and the interplay between drug and bone regeneration should be also assessed in vivo”.

Regarding the dug that was finally incorporated into the fabricated scaffold, as reported in section 3.2, ‘’PCL-TCH scaffolds demonstrated a very high DEE% of 95 % ± 3.6 %, and a measured drug loading of 0.95% (± 0.04%)’’. Thus, the final fabricated scaffold contained 0.95% w/w TCH.

  1. Tetracycline hydrochloride is commonly known to be thermally degraded and has been shown to be unstable (Ref: Y. Wu and R. Fassihi, Stability of metronidazole, tetracycline HCl and famotidine alone and in combination, International Journal of Pharmaceutics, 2005, 290, 1-2, 1-13.) However, the authors processed it at temperatures as high as 140oC to fabricate the composite scaffold. Please provide evidence for the stability of the drug in this regard.

We would like to thank the Reviewer for his/her comment and for citing the above reference. However, after carefully reading the above paper, we understand that Wu & Fassihi examined the stability of tetracycline hydrochloride under various storage conditions –either in solid or liquid state– for a prolonged period of time (15–90 days). In our study, we took into account the stability of TCH during printing. In our case, the heating of PCL and tetracycline hydrochloride at 140 ºC takes place for a short amount of time (<30 min) to produce the desired 3D-scaffolds. In order to ensure that tetracycline hydrochloride remains stable at 140 ºC, we studied in advance its thermal behavior via thermogravimetric analysis, which is presented in Fig.1C. The results showed that the drug remains stable up to 225 ºC, when thermal decomposition begins.

  1. The authors stated that there was no direct prior research showing that osteoblast differentiation was activated by tetracycline hydrochloride. To reinforce this, please add information about prior research to the discussion section. Or, strengthening data is needed to prove it experimentally. (Example: qRT-PCR data using various specific genes related to osteogenesis that can confirm osteogenic diffrentiation, etc.)

References

- S. Farzamfar et al., Tetracycline hydrochloride-containing poly (e-caprolactone)/poly lactic acidscaffold for bone tissue engineering application:in vitroandin vivostudy, Tetracycline, International Journal of Polymeric Materials and Polymeric Biomaterials, 2019, 68, 8, 472–479

- E. Dayaghi et al., Magnesium-zinc scaffold loaded with tetracycline for tissue engineering application: In vitro cell biology and antibacterial activity assessment, Materials Science and Engineering: C, 2019, 102, 53-65.

- W. Wang et al., Effect of tetracycline hydrochloride application on dental pulp stem cell metabolism–booster or obstacle for tissue engineering?, Frontiers in Pharmacology, 2023, 14, 1277075.

As suggested by the reviewer, we tried to enforce the discussion section by adding new text and by citing additional literature data as indicated.

  1. Surface and cross-sectional SEM images of the control group (PCL) and experimental group (PCL/TCH) should be added to the manuscript.

Surface of both scaffold variants were added in figure 3. Additionally, the legend in figure 3 was changed accordingly. Appropriate changes were also done within text.

Cross sections of the scaffolds were skipped, because our prior experience showed that when we have tried to cut these kinds of scaffolds, either before or after fixation, resulted to the destruction of the internal scaffold’s architecture. Any useful details that can be observed under SEM were effortless. For clarity, we endorse an example of own previous cross section images out of PCL scaffolds.

  1. The resolution of Figure 2A and Figure 7 images must be improved as well as the high magnification images must be provided.

Figure 2 has been reorganized. The resolution has been increased accordingly and high magnification images of representative ROI’s has been provided. The figure legend and the text has been changed accordingly. Figure 7 has been reorganized, and increased resolution has been adapted to the figure. The figure legend and the text has been changed accordingly.

Reviewer 2 Report

Comments and Suggestions for Authors

Line 161, delete cell infiltration.

Line 162, osteogenic not osteoinductive. Why did the osteogenic medium was added after 7 days? Usually is after 1 or two days since seeding.

Delete "An effort was made...."in the collagen quantification.

Line 241, swith In brief with Briefly.

Figure 2A must be changed in order to have more organized photos.

2d controls should have been carried out for all the experiments.

Figure of paragraph 3.6 should be switched with a graph or a table showing the area of inhibition.

Results of collagen contents should have been normalised against DNA amount.

Alp results should have been normalised versus DNA amount.

Figure 7 should have images with all the same size. 

Comments on the Quality of English Language

Moderate English editing.

Author Response

Reviewer 2

We would like to thank the Reviewer for the time devoted to review our manuscript and for hir/her appreciation of our work, the valuable comments, as well as constructive criticism. Following his/her valuable suggestions, the manuscript has been improved and all the points raised are answered one by one.

Line 161, delete cell infiltration.

As suggested by the reviewer, it has been deleted and we have corrected the specific point.

Line 162, osteogenic not osteoinductive. Why did the osteogenic medium was added after 7 days? Usually is after 1 or two days since seeding.

In our study, osteoprecursor cells were used as a model to evaluate osteoblast differentiation, as it is well known that these cells proliferate and can differentiate to more mature osteoblasts.[49, 50] In this regard we used complete medium the first 7 days of cell culture to gain a satisfactory amount of cells within 3D scaffolds, before adding osteogenic medium that enhances cell differentiation.

Delete "An effort was made...."in the collagen quantification.

We rephrased the sentence. Please, now read in lines 229-231

“Total amount of collagen per scaffold was quantified by dissolving the bounded dye.”

Line 241, switch In brief with Briefly.

We have accordingly corrected the specific point.

Figure 2A must be changed in order to have more organized photos.

As suggested by the reviewer, figure 2 has been reorganized. Additionally, high magnification images of representative ROI’s have been added and the resolution of the images have been increased. The figure legend and the text has been changed accordingly

2d controls should have been carried out for all the experiments.

We would like to thank the reviewer for the comment. Our primary objective was to evaluate and compare the performance of treated and untreated 3D scaffolds. Introducing a 2D control group would bring out key differences regarding the proliferation and differentiation rate between 2D and 3D cultures. By intentionally omitting a 2D cell culture control, we aimed to focus our experimental design, statistical analysis, and discussion to better capture the differences between the drug-loaded and drug-free 3D scaffolds.

Figure of paragraph 3.6 should be switched with a graph or a table showing the area of inhibition.

We thank the reviewer for his/her suggestion. Instead of adding a table with the results highlighting the areas of inhibition, we preferred to add these results in the figure legend, so that to enhance clarity and readability. We have kept the figure, as we believe that it better represents the testing technique against S. aureus LMG 8224.

Results of collagen contents should have been normalised against DNA amount.

We would like to thank the reviewer for his/her suggestion. Unfortunately, we did not have the infrastructure (nor access) to run a RT-qPCR analysis for finding specific types of collagen within the newly formed tissue on our scaffolds. However, biochemical assay with picrosirious red is well known since picrosirious red binds to collagen and used in routine-based bone histological sections, as also to other tissues containing collagen. We used it as an alternative method to our 3D-scaffolds, and we have managed to quantify total collagen amount after the dye was dissolved from the entire scaffold, rather than to perform histological slides that will partially demonstrate collagen content.

Alp results should have been normalised versus DNA amount.

We do understand the point of the reviewer, however according to the literature (Methods Protoc. 2020;3(2):30. doi: 10.3390/mps3020030), normalizing ALP content to the total protein content or dsDNA concentration of the examined samples are both acceptable and well-documented methods applied. In this regard and based on the ALP kit's instructions (Sigma-Aldrich, USA), we proceeded to normalize the ALP values based on protein content.

Figure 7 should have images with all the same size. 

Figure 7 has been reorganized to and now the images are the same size. The figure legend and the text has been changed accordingly.

Reviewer 3 Report

Comments and Suggestions for Authors

In this manuscript, the researchers have developed a new type of 3D-printed antibacterial scaffold that could help to regenerate bone tissue in severe periodontitis. The scaffold is made of a material called poly(ε-caprolactone) (PCL) and loaded with tetracycline hydrochloride (TCH), an antibiotic. The scaffold was evaluated on osteoprecursor cells (MC3T3-E1), which are cells that can develop into bone cells. The results showed that the scaffold promoted the growth of collagen, alkaline phosphatase (ALP) activity, and mineralized bone tissue. The scaffold also inhibited the growth of Staphylococcus aureus, a bacterium that is often associated with periodontitis. The researchers suggest that the scaffold could be used as an alternative to traditional implants for treating severe periodontitis.

Dear authors, it was a pleasure to read you're interesting manuscript there is really not much to suggest for improvements in some parts of your manuscript.

In general, your manuscript would benefit from more up to date citations for the current research in these fields, for example in various types of scaffolds.

Lines 132 and 135 are two compacted and the formulas should have more space 132 and 135 are two compacted and the formulas should have more space.

Table 1 is not according to authors instructions for IJMS. The table shall be remade according to the instructions of the authors of this Journal.

Figure #1 he's using a different font than prescribed there is difference between words temperature on the left and right side of the image. Also figures 2A and B use titles with extra-large font and legends don’t meet the required criteria.

Figure #7 is not well readable and is a bit confusing, try to improve,

The discussion chapter is extremely long, and I strongly suggest creating a chapter Conclusions that would summarize the findings for your readers.

Here are four suggestions for improvements to the Discussion chapter of your manuscript:

1.      Discuss further the possible pathways through which PCL-TCH scaffold aid-bone formation. In prolonged cultures, TCH could block cell proliferation while leaving them viable. You further add that increased release of TCH from the scaffold could promote osteogenic differentiation. Nevertheless, there is a need for further elaboration of the molecular and cellular processes in order to explain how these effects occur.

2.      Identify the weaknesses of the study and highlight potential areas for further investigations. As for future research, one would need to optimize the concentration of TCH into the scaffolds and evaluate how the interaction between the drug and bone regeneration occurs in vivo. At the same time, I am producing the anti-inflammatories laden scaffolds, which This chapter can benefit greatly by discussing limitations and propositions for further research in greater depth.

3.      Please discuss the potential of TCH-loaded 3D-printed scaffolds to improve other 3D-printed scaffolds for bone regeneration - for example fabrication of novel 3D printed  hydroxyapatite scaffolds - Fabrication and In Vitro Characterization of Novel Hydroxyapatite Scaffolds 3D Printed Using Polyvinyl Alcohol as a Thermoplastic Binder DOI  https://doi.org/10.3390/ijms232314870 And discuss here the need for further research to assess the safety and efficacy of 3D-printed scaffolds for bone regeneration in periodontal diseases in vivo as well as synergistic effects of TCH and other bioactive agents that could be loaded into 3D-printed scaffolds for bone regeneration in periodontal diseases.

4.      Ensure the unity of the discussion chapter and strengthen its entirety. To have a better version of the discussion chapter, all the sections should be arranged logically. For instance, you may elaborate on the possible benefits of PCL-TCH scaffolds compared to other therapies in treating severe periodontitis. At this point, you can elaborate on the specific results of their research such as the effect of TCH on population, differentiation, and bone formation. Finally, you could talk about the weaknesses of the study as well as proposed recommendations on potential areas for future studies.

Otherwise I consider the paper useful for the clinical field and I see it woth publication after the adjustments..

Comments on the Quality of English Language

is fine

Author Response

Reviewer 3

We would like to thank the Reviewer for the time devoted to review our manuscript and for hir/her appreciation of our work, the valuable comments, as well as constructive criticism. Following his/her valuable suggestions, the manuscript has been improved and all the points raised are answered one by one.

Comments and Suggestions for Authors

In this manuscript, the researchers have developed a new type of 3D-printed antibacterial scaffold that could help to regenerate bone tissue in severe periodontitis. The scaffold is made of a material called poly(ε-caprolactone) (PCL) and loaded with tetracycline hydrochloride (TCH), an antibiotic. The scaffold was evaluated on osteoprecursor cells (MC3T3-E1), which are cells that can develop into bone cells. The results showed that the scaffold promoted the growth of collagen, alkaline phosphatase (ALP) activity, and mineralized bone tissue. The scaffold also inhibited the growth of Staphylococcus aureus, a bacterium that is often associated with periodontitis. The researchers suggest that the scaffold could be used as an alternative to traditional implants for treating severe periodontitis.

Dear authors, it was a pleasure to read you're interesting manuscript there is really not much to suggest for improvements in some parts of your manuscript.

In general, your manuscript would benefit from more up to date citations for the current research in these fields, for example in various types of scaffolds.

Lines 132 and 135 are two compacted and the formulas should have more space 132 and 135 are two compacted and the formulas should have more space.

We have accordingly corrected the specific point based on the reviewer’s comment.

Table 1 is not according to authors instructions for IJMS. The table shall be remade according to the instructions of the authors of this Journal.

We have accordingly corrected the Table, based on the Journal instructions.

Figure #1 he's using a different font than prescribed there is difference between words temperature on the left and right side of the image. Also figures 2A and B use titles with extra-large font and legends don’t meet the required criteria.

Figure 1 has changed accordingly. Figure 2 has been reorganized. The fonts and the legends were corrected. Additionally, high magnification images of representative ROI’s have been provided.

Figure #7 is not well readable and is a bit confusing, try to improve,

As suggested by the reviewer figure 7 has been reorganized. The resolution has been increased accordingly to make images more clear The figure legend and the text has been changed accordingly.

The discussion chapter is extremely long, and I strongly suggest creating a chapter Conclusions that would summarize the findings for your readers.

We would like to thank the Reviewer for the suggestion. In our draft manuscript before submission, we had created a conclusion section, but intentionally we decided to integrate our conclusions within the paragraph containing limitations and future perspectives of our study, aiming to provide a more concise and comprehensive overview for the readers.  

Here are four suggestions for improvements to the Discussion chapter of your manuscript:

We would like to thank the Reviewer for his/her kind suggestions to improve clarity and readability of the discussion section. We have tried to incorporate all of his/her comments to improve the discussion chapter.

  1. Discuss further the possible pathways through which PCL-TCH scaffold aid-bone formation. In prolonged cultures, TCH could block cell proliferation while leaving them viable. You further add that increased release of TCH from the scaffold could promote osteogenic differentiation. Nevertheless, there is a need for further elaboration of the molecular and cellular processes in order to explain how these effects occur.

We have accordingly tried to improve different parts of the discussion chapter, regarding the possible osteoinductive effect of TCH.

Lines 509 - 519 of the revised manuscript:

“Pertaining to cell differentiation, bone formation and mineralisation on 3D scaffolds loaded with tetracycline, few data exist in the literature, supporting that tetracyclines at low concentrations may facilitate osteogenesis. [43] Nevertheless, several tentative mechanisms have been proposed through which tetracycline antibiotics modulate osteogenic differentiation. For example, tetracycline hydrochloride (at 1 μg/mL) was found to promote in vitro osteogenesis in human bone marrow stem cells by upregulating the Wnt signalling pathway. [44] Also, this class of compounds seems to inhibit the activity of various matrix metalloproteinases (MMPs), therefore impeding osteoclast activity that leads to osteopenia and osteoporosis. [43, 45-48]

And lines 550-552:

“Further studies are required to fully explore their potential and shed light on the cellular and molecular mechanisms that govern osteogenesis, in the presence of TCH-containing scaffolds”

  1. Identify the weaknesses of the study and highlight potential areas for further investigations. As for future research, one would need to optimize the concentration of TCH into the scaffolds and evaluate how the interaction between the drug and bone regeneration occurs in vivo. At the same time, I am producing the anti-inflammatories laden scaffolds, which This chapter can benefit greatly by discussing limitations and propositions for further research in greater depth.

As kindly proposed by the Reviewer, we tried to further improve the specific part in the discussion chapter.

Lines 559-562 of the revised manuscript:

“To this end, it will be interesting to study the synergistic effect of both anti-inflammatory and antibiotic compounds on bone remodelling and bone regeneration and investigate their release kinetics from such scaffold matrices.”

In general, we believe that we have already notified and discussed the possible limitations and future perspectives of our study, as well as future steps, as it is mentioned in the manuscript:

“In this study, we demonstrated that tetracycline-loaded PCL scaffolds can promote cell differentiation and bone formation at subsequent stages compared to neat PCL scaffolds, when cultured in vitro for 21 days. This research brings promising results in the field of applying tissue engineering API-loaded scaffolds to periodontal diseases. Further studies are required to fully explore their potential and shed light on the cellular and molecular mechanisms that govern osteogenesis, in the presence of TCH-containing scaffolds. Additionally, the concentration of TCH within scaffolds should be further optimized in terms of cell proliferation and differentiation and the interplay between drug and bone regeneration should be also assessed in vivo in long-term studies. This will allow us to evaluate the safety and the efficacy of TCH-loaded scaffolds, which is an important preclinical step before transitioning to clinical trials. Taking also into account that bacteria and inflammation co-exist in periodontitis, the development of anti-inflammatory agent-loaded scaffolds is also in progress in our laboratory. To this end, it will be interesting to study the synergistic effect of both anti-inflammatory and antibiotic compounds on bone remodelling and bone regeneration and investigate their release kinetics from such scaffold matrices.”

  1. Please discuss the potential of TCH-loaded 3D-printed scaffolds to improve other 3D-printed scaffolds for bone regeneration - for example fabrication of novel 3D printed hydroxyapatite scaffolds - Fabrication and In Vitro Characterization of Novel Hydroxyapatite Scaffolds 3D Printed Using Polyvinyl Alcohol as a Thermoplastic Binder DOI https://doi.org/10.3390/ijms232314870. And discuss here the need for further research to assess the safety and efficacy of 3D-printed scaffolds for bone regeneration in periodontal diseases in vivo as well as synergistic effects of TCH and other bioactive agents that could be loaded into 3D-printed scaffolds for bone regeneration in periodontal diseases.

We would like to thank the Reviewer for kindly referencing the above study. We have accordingly emphasized in the discussion section the need for further research to evaluate the safety and efficacy of 3D-scaffolds for bone regeneration.

Lines 555-557 of the revised manuscript:

“This will allow us to evaluate the safety and the efficacy of TCH-loaded scaffolds, which is an important preclinical step before transitioning to clinical trials.”

  1. Ensure the unity of the discussion chapter and strengthen its entirety. To have a better version of the discussion chapter, all the sections should be arranged logically. For instance, you may elaborate on the possible benefits of PCL-TCH scaffolds compared to other therapies in treating severe periodontitis. At this point, you can elaborate on the specific results of their research such as the effect of TCH on population, differentiation, and bone formation. Finally, you could talk about the weaknesses of the study as well as proposed recommendations on potential areas for future studies.

We built up the discussion part in a logical way so that to critically comment on all our results, in the order they are presented in the results’ section. Following that, we concluded on highlighting the limitations of our study, future studies needed and perspective of our work. As kindly suggested, we have furthermore accordingly enhanced the discussion chapter by adding more text and stressing the potential of our findings as an alternative treatment for severe periodontitis.

Otherwise I consider the paper useful for the clinical field and I see it worth publication after the adjustments.

Once again, we thank the Reviewer for the appreciation of our work, the positive comments on our research and for the time devoted to improve our manuscript.

Round 2

Reviewer 1 Report

Comments and Suggestions for Authors

I suggest publishing the manuscript as it has been thoroughly revised and is in good condition.

Author Response

We would like to thank the Reviewer for the time devoted to review our manuscript and for hir/her acceptance.

Reviewer 2 Report

Comments and Suggestions for Authors

The authors should do a further revision of the manuscript making sure the figures are clear and cited in the text (e.g. Fig 6 and Fig.7). For Figure 6, if you decide to show the photos taken at day 7 for PCL, then the authors must show the same time point for the other materials. Some sentences need extensive English revision, such as line 321, 324 and 509

I have asked the authors to normalise the total amount of collagen and ALP against DNA. The authors replied they didn't have the facility to run RT-qPCR. The DNA quantification can be easily done using DNA staining such as SYBR green (commercial kits are available) and reading the fluorescence with a plate reader. Alternatively, some papers use show the collagen and ALP versus total proteins (which is not ideal, but acceptable). Again protein quantification can be done using colorimetric kits and a plate reader. Unfortunately, this is a serious flaw of this paper that needs to be addressed. 

Delete "basic" from "basic marker of early osteogenesis"

In the legends, please add which statistical analysis has been carried out. It is clearer for the reader to have all the basic information in the legends.

Paragraph 3.5 is incomplete: despite showing figures for time point 21, nothing is said about it. From the images, it isn't clear that the cells on PCL-TCH formed sheets of tissue. 

Why do the metabolic activity decreases from day 1 to 7, especially for PCL scaffolds? Elaborate this.

Collagen production is not a marker of osteoblastic activity. You didn't look into the production of collagen I, which is a better marker for osteoblastic activity.

It is interesting that ALP activity was higher at day 21 than at day 7, when ALP is usually considered as an early marker of osteogenesis. Elaborate this.

Comments on the Quality of English Language

Moderate English editing

Author Response

Reviewer 2 - Round 2

Once more, we would like to thank the Reviewer for the time devoted to review our manuscript and the valuable comments for its improvement. Following his/her suggestions, the manuscript has been amended and all the points raised are answered one by one. Furthermore, revisions of the main text of the manuscript are highlighted in the -already revised version- in blue.

The authors should do a further revision of the manuscript making sure the figures are clear and cited in the text (e.g. Fig 6 and Fig.7). For Figure 6, if you decide to show the photos taken at day 7 for PCL, then the authors must show the same time point for the other materials. Some sentences need extensive English revision, such as line 321, 324 and 509

Following the suggestions of the reviewer, we have changed figure 7, presenting images of both scaffolds only from days 14 and 21 with appropriate renumbering, since images from day 7 did not display any useful information. Of course, the results for day 7 are still mentioned in the text in paragraph 3.7.3, where we state “No fluorescence signal was detected on PCL-TCH scaffolds on day 7, while minor signs of green stain dots were observed on PCL scaffolds (data not shown).”. Further, all our figures are clearly cited within the text.

As also suggested by the reviewer, we have changed the sentences in lines 321, 324 and 509, now found in lines 321-326 and 534 in the revised manuscript.

I have asked the authors to normalise the total amount of collagen and ALP against DNA. The authors replied they didn't have the facility to run RT-qPCR. The DNA quantification can be easily done using DNA staining such as SYBR green (commercial kits are available) and reading the fluorescence with a plate reader. Alternatively, some papers use show the collagen and ALP versus total proteins (which is not ideal, but acceptable). Again protein quantification can be done using colorimetric kits and a plate reader. Unfortunately, this is a serious flaw of this paper that needs to be addressed.

We fully agree with the reviewer that both Collagen and ALP can be more precisely presented when normalised against DNA. As an alternative, the reviewer suggested the use of total proteins and therefore we proceeded to normalize the ALP to the total protein content.

It is worth mentioning that previous studies have documented scaffold staining with Picrosirius-red, with their results expressing the amount of collagen present on their constructs (https://doi.org/10.1089%2Ften.tec.2013.0041,  https://doi.org/10.1002%2Fjbm.a.30122). Having in mind these studies, we tried to use PS-red stain and we did not know whether this preliminary technique will work for our scaffolds and to our experimental settings. Due to the preliminary nature of our test, and after observing that the stain worked on our constructs (figure 5.A), we tried to quantify collagen concentration by measuring the absorbance values, as stated in our text. To this end and to avoid any confusion to the readers, we have changed our headings on figure 5.A “Scaffolds stained with Picrosirius Red” and on figure 5.B to “Amount of collagen found on 3D scaffolds”. Additionally, we have rephrased accordingly within the manuscript.

However, we certainly agree with the reviewer that it is more precise to express collagen versus DNA, and we are willing to invest οn SYBR green kits for our future experiments. Thank you again for your useful suggestion.

Delete "basic" from "basic marker of early osteogenesis"

As suggested by the reviewer, it has been deleted and we have corrected the specific point.

In the legends, please add which statistical analysis has been carried out. It is clearer for the reader to have all the basic information in the legends.

As suggested by the reviewer, we have added the statistical analysis method carried out to the corresponding figure legend for each assessment.

Paragraph 3.5 is incomplete: despite showing figures for time point 21, nothing is said about it. From the images, it isn't clear that the cells on PCL-TCH formed sheets of tissue.

We thank the reviewer for his/her suggestion. We have now added new text for SEM observations regarding day 21. Additionally, we have rephrased our observations based on our figures to enhance readability. Please now read lines 362-370 of the revised manuscript.

Why do the metabolic activity decreases from day 1 to 7, especially for PCL scaffolds? Elaborate this.

Cell activity was measured via the resazurin based assay. From our results, PCL scaffold seemed to have a two-fold increase at the first 3 days, as we mentioned in our text in lines 326-328 “Cells that were cultured on PCL scaffolds, showed approximately a two-fold increase (1.26 ± 0.05) of their metabolic activity on day 3, compared to the metabolic activity observed on day 1 (0.59 ± 0.08).” Additionally, we found that cells on PCL scaffolds maintained their metabolic activity on day 5 followed by a minor reduction on day 7.

  1. This can be attributed to specific geometrical topologies within the scaffolds’ structure, where cells adhered, proliferated and reached confluency.
  2. Furthermore, deviation in growth kinetics can originate from the scaffold material, the media type, culturing conditions, the number of prior passages, cell seeding density, cell counting assay and the selected time points of each measurement. Based on the aforementioned reasons, studies using similar cells (MC3T3) with variations on their proliferative phase, have been previously reported (e.g., https://doi.org/10.1002/jbmr.5650070613, https://doi.org/10.1111/j.1365-2184.1996.tb00980.x,), or for 3D scaffolds ( https://doi.org/10.1016/j.biomaterials.2009.09.063)
  3. Immature osteoblasts are known to follow a specific pathway towards their differentiation to more mature osteoblasts and finally to osteocytes. Their proliferation rate decreases while they start expressing osteogenic markers such as alkaline phosphatase secreted by early matured osteoblasts, (https://doi.org/10.1186/s13287-018-0995-x). From our results, we observed a small ALP expression on day 7. This can be another logical explanation to the results based on the “resazurin assay”, that showed a “minor decrease in day 7” on PCL scaffolds.

Based on our results, PCL-TCH scaffold seemed to have an insignificant small increase in day 3, and afterwards cell activity was hindered by approximately 57 % and remained in the same condition till day 7. We have also discussed these findings in the discussion section: “Cell proliferation was inhibited by PCL-TCH scaffolds, from day 3 to day 7 of culture, which agrees with the study of Ferreira et al. that combined TCH with polydioxanone (PDS).[40] However, it is notable that PCL-TCH scaffolds exhibited an insignificant amount of dead cells on day 7, comparable to those found in PCL scaffolds, observed by the CLSM analysis in Figure 2.A. Therefore, it appears that tetracycline antibiotics act as a barrier, hindering cell proliferation but without altering their viability on prolonged cultures. In agreement, Park et al. reported that cell viability resulted in insignificant differences when preosteoblasts cultured without or with tetracycline, in the range of 0.1 and 1.0 μM.[41] Yet, the same researcher in another study, exclaimed that higher doses of tetracycline in the range of 100–1000 μM can yield a negative effect on cell viability.[42]

Collagen production is not a marker of osteoblastic activity. You didn't look into the production of collagen I, which is a better marker for osteoblastic activity.

We agree with the reviewer that collagen is not a marker of osteoblastic activity, and thus we rephrased our sentence in lines 536-538, to “…was found on both PCL and PCL-TCH scaffolds at the end of the culture, confirming the production of the basic organic structural matrix”.

It is interesting that ALP activity was higher at day 21 than at day 7, when ALP is usually considered as an early marker of osteogenesis. Elaborate this

We agree with the reviewer that ALP is usually considered as an early marker of osteogenesis.

During the pre-osteoblastic cycle, ALP is expressed when accumulated inorganic phosphate groups (Pi) are generated. Pi groups with calcium ions further deposit amorphous calcium phosphate or hydroxyapatite crystals within the secreted ECM (https://doi.org/10.1021/acsomega.1c07225, doi: 10.1016/j.gene.2020.144855). This is considered as an early phase marker of bone tissue formation, while at later phases ALP expression reaches a peak (plateau) value and finally decreases indicating the mineralisation of ECM (a. https://doi.org/10.1016/j.biomaterials.2005.05.046, b. https://doi.org/10.1002/(sici)1097-4644(19980201)68:2%3C269::aid-jcb13%3E3.0.co;2-a,    c. https://doi.org/10.1002/(sici)1097-4644(19960616)61:4%3c609::aid-jcb15%3e3.0.co;2-a )

Additionally, pre-osteoblast differentiation covers various stages towards maturation and osteoblastic lineage development. ALP can be found highly expressed on both immature osteoblasts and to more mature osteoblasts (as presented in figure 1 below), while in the same time the mineralisation phase can also begin (https://doi.org/10.3390/ijms22062851).

Figure 1 retrieved from Int. J. Mol. Sci. 2021, 22(6), 2851; https://doi.org/10.3390/ijms22062851

Based on the above, our PCL scaffolds exhibited a continuously increasing ALP activity suggesting a state of continuous osteoblast maturation, while calcein (figure 7 of our manuscript) showed early stages of ECM maturation and mineralisation. This can be attributed to the fact that when preosteoblasts become more differentiated, these two different phases can co-exist.

Reviewer 3 Report

Comments and Suggestions for Authors

The authors have not paid enough attention to my comments and have treated them very superficially. Nor have they added any references to the added statements.

Comments on the Quality of English Language

is ok

Author Response

The authors have not paid enough attention to my comments and have treated them very superficially. Nor have they added any references to the added statements.

Based on the comment of Reviewer 3, we would like to state that our responses to his Round 1 comments have been misunderstood. We truly appreciate his/her input and the time he/she devoted to improve our manuscript, taking into careful consideration all the statements he/she made. In that respect, we have thoroughly re-read his/her suggestions and tried to incorporate them to our manuscript. Specifically, we have completely re-structured the Discussion chapter by i) arranging the different sections in a more rational manner, ii) adding more text and references where needed and iii) generally, elaborating more on the various key points that were raised by Reviewer 3, during the Review Report Round 1. We consider now that the Discussion part of our manuscript has been properly modified and significantly strengthened, according to the suggestions of Reviewer 3. All the discussion ids now highlighted in blue colour.

Round 3

Reviewer 2 Report

Comments and Suggestions for Authors

The authors have replied to my comments.

Comments on the Quality of English Language

Minor English editing

Reviewer 3 Report

Comments and Suggestions for Authors

Thank you for response to majority of my remarks.

Comments on the Quality of English Language

is fine